# Biomolecules of Muscle Fatigue in Metabolic Myopathies

**DOI:** 10.3390/biom14010050

**Published:** 2023-12-30

**Authors:** Erika Schirinzi, Giulia Ricci, Francesca Torri, Michelangelo Mancuso, Gabriele Siciliano

**Affiliations:** Department of Clinical and Experimental Medicine, Neurological Clinic, University of Pisa, 56126 Pisa, Italy

**Keywords:** muscle fatigue, biomolecules, metabolic myopathies

## Abstract

Metabolic myopathies are a group of genetic disorders that affect the normal functioning of muscles due to abnormalities in metabolic pathways. These conditions result in impaired energy production and utilization within muscle cells, leading to limitations in muscle function with concomitant occurrence of related signs and symptoms, among which fatigue is one of the most frequently reported. Understanding the underlying molecular mechanisms of muscle fatigue in these conditions is challenging for the development of an effective diagnostic and prognostic approach to test targeted therapeutic interventions. This paper outlines the key biomolecules involved in muscle fatigue in metabolic myopathies, including energy substrates, enzymes, ion channels, and signaling molecules. Potential future research directions in this field are also discussed.

## 1. Introduction

### 1.1. Determinants of Muscle Contractions and Fatigue Definition

Muscle fatigue (MF) is one of the most frequently encountered symptoms in clinical practice. In the general population, a condition of fatigue is reported with a prevalence from 5 to 45% as temporary and over 10% as persistent [1]. In turn, neurological diseases, including neuromuscular ones, present the highest prevalence of the symptom [1,2]. It is worth noting that MF is a complex phenomenon. Nevertheless, a large number of definitions are widely accepted: sometimes fatigue is identified by patients through descriptive metonymies such as “exhaustion, fatigability, weakness, lack of vigor” [3]. Based on physiological mechanisms, fatigue is defined as a decline or inability to maintain over time an adequate level of force generated by the muscular contractile apparatus [4]. This definition highlights a link between fatigue and the temporal dimension of continuous or repeated motor activity, rather than contractile insufficiency at rest, which is semiologically identified as hyposthenia or strength deficit [4]. In pathological conditions, such as neuromuscular diseases, skeletal muscle hyposthenia and fatigue strictly coexist, conditioning the muscle performance at various extents.

Voluntary muscle contraction is the result of changes in the sequence of physiological processes both in the nervous system and in muscles, which can affect the quantitative and qualitative aspects of the contraction itself [5].

The process of muscle contraction begins with the activation of motor neurons in the brain or spinal cord. These motor neurons generate action potentials that propagate towards the neuromuscular junctions that release acetylcholine neurotransmitter to the muscle fibers. Subsequently, the acetylcholine binds to its receptors on the muscle fiber’s membrane, causing depolarization by changes in the membrane’s permeability to sodium ions. The action potential spreads along the surface of the muscle fiber and through the transverse tubules and induces the sarcoplasmic reticulum to release Ca^2+^ into the sarcoplasm [6]. Actin and myosin are the main components of sarcomere, the contractile unit of the muscle. The myosin heads bind to the actin filaments and pull them towards the center. As a result, the sarcomere within the muscle fiber shortens and the entire muscle fiber contracts. This process is dependent on the immediate availability of creatine phosphate and the activity of adenosine triphosphatase (ATPase) enzyme, which hydrolyzes ATP into ADP (adenosine diphosphate) and inorganic phosphate, providing the energy necessary for the cross-bridge cycling between actin and myosin [7]. At the end of this process, the myosin heads detach from actin to allow for the next contraction cycle. Once the action potential runs out, Ca^2+^ is re-uptaken into the sarcoplasmic reticulum. The low Ca^2+^ concentration in the sarcoplasm leads to the inhibition of myosin-actin cross-bridge formation, and the muscle relaxes [4,7]. The precise coordination of these events allows the muscles to generate and maintain force and any impairment in this molecular pathway can affect muscle structure and contraction, as well as force production and endurance. It should be mentioned that the response of skeletal muscle fibers depends on what type of muscle fiber, slow-twitch or fast-twitch, is activated in relation to the motor task required [8]. Slow-twitch or type I fibers predominantly rely on aerobic metabolism; therefore, they have a rich supply of mitochondria for generating ATP through oxidative phosphorylation and contain a high concentration of myoglobin that stores oxygen within muscle cells. This allows them to sustain contractions for longer durations but, compared to fast-twitch fibers, they generate less force. Fast-twitch or type II fibers rely on anaerobic metabolism or are oxidative glycolitic and have a higher myosin ATPase activity, which enables fast and strong muscle contractions; these fibers are better suited for short bursts of intense activity because they generate energy quickly, but only for a short time [5]. In skeletal muscles, a mixture of both slow-twitch and fast-twitch fibers coexists, with varying proportions depending on their specific function, genetics, and training factors.

In metabolic MF condition, either in normal or, at a greater extent, in metabolic myopathies, the demand for energy substrates (such as ATP and its sources) exceeds the rate at which they can be produced. These conditions can also lead to abnormal accumulation of intermediate byproducts of interrupted or alternative biochemical pathways, which may serve as indicators of the cause of imbalance in the energy production system. Whichever the origin, the metabolic buildup can contribute to fatigue as a temporary and reversible event related to exhaustion, likely in combination with additional secondary effects of the causative metabolic defect, such as sarcomeric loss or mitochondrial population remodeling related to under training/deconditioning. In particular, the accumulation of lactic acid, hydrogen ions, and inorganic phosphate can interfere with muscle function, impairing Ca^2+^ release and reuptake, and disrupting the excitation–contraction coupling process [9,10]. In this case, adequate rest and recovery periods between workouts allow for the replenishment of the energy supplies and the repair of muscle tissue. In addition, repeated muscle contractions and the associated mechanical stress can lead to structural muscle damage. This occurs in healthy conditions and is exacerbated in disease conditions, such as myopathies, in which the innate robustness of muscle fibers is damaged and the symptoms become chronic [4,7]. The mechanical damage, in turn, triggers an inflammatory response, which contributes to sustain the phenomenon. Finally, the accumulation in the central nervous system (CNS) of metabolic byproducts generated during prolonged or intense exercise, such as lactate and ammonia, or inflammatory mediators such as interleukin-1 and -6, as well as tumor necrosis factor alpha has also been linked to fatigue [10,11]. These byproducts can directly affect the CNS and modulate neurotransmitter activity, interfering with the normal neuronal function and contributing to the feeling of exhaustion, reduced motivation, and decreased mental and physical performance [12]. It has been suggested also that alterations in neurotransmitter balance and their activity, such as changes in the availability of serotonin, dopamine, and norepinephrine, within the CNS can influence fatigue perception [13]. In this context, the additional contribution of psychological factors, such as stress, anxiety, and mood disturbances, which in turn can be related to neurotransmitter imbalance, has to be taken into account [14,15]. Nevertheless, prolonged muscle activity can result in decreased motor unit recruitment and firing rates, leading to reduced muscle force production with consequent neural drive impairment to activate and coordinate muscle fibers. However, while by the end of the 19th century the distinction, theorized by Angelo Mosso [16] between the decrease in strength and perceived feeling of fatigue following prolonged muscle activity is still valid [17], a dichotomic split between “peripheral” and “central” contribution in explaining the muscle fatigue phenomenon is very difficult.

In metabolic myopathies, exertional or post-exertional fatigue can be associated with satellite symptoms such as muscle contractures, pain, myoglobinuria, due to a depletion in ATP stocks and myofibril damage. However, paradoxically, there are conditions in which exercise can improve some motor performances, such as the “second wind” phenomenon in McArdle disease for the anaplerotic effects of physical exercise [18].

### 1.2. Classification of Peripheral Fatigue Determinants

As fatigue is an epiphenomenon related to complex pathophysiological and structural alterations of the muscle, its occurrence is likely affected by biomolecular imbalances. Therefore, studying biomolecules in the context of muscle fatigue is crucial for understanding the underlying mechanisms and processes involved in this phenomenon. However, in an attempt to classify these determining factors, different points of view can be considered.

#### 1.2.1. Classification by Pathophysiology/Pathogenesis

Pathophysiology/pathogenesis classification is based on the alteration of neuromuscular junction, inability of electro-mechanical coupling [19], dysregulation of biomolecules involved in muscle contraction and relaxation [20] or in energy production and metabolite accumulation [21], and myofiber transmembrane ion flow impairment [19]. Moreover, after exercise or injuries, several biomolecules intervene in muscle restoring by satellite cells activation and myoblast differentiation, extracellular matrix rearrangement, inflammation response, and angiogenesis [22,23]. Over time, the newly formed muscle fibers undergo remodeling and maturation to optimize their structure and function. This process may take from several weeks to months, during which the muscle gradually recovers its strength and function [24].

#### 1.2.2. Muscle Contraction Energy Source

The continuous supply of energy substrates is fundamental to underpin skeletal muscle contraction during exercise lasting from seconds to several hours. In particular, glucose and fatty acids are the two primary energy sources [25,26].

(i)Glucose

Glucose is a sugar molecule obtained from the breakdown of carbohydrates in the diet or through glycogen stores in the liver and muscles. Glucose enters the muscle cells through glucose transporters (GLUTs) and undergoes a series of metabolic reactions to generate ATP [27]. Through the glycolysis process, glucose is initially converted to pyruvate in the cytoplasm, to generate a small amount of ATP and NADH. Pyruvate metabolism depends on the energy demands and availability of oxygen. In aerobic conditions, pyruvate enters the mitochondria and undergoes the citric acid cycle (Krebs cycle) to generate ATP through oxidative phosphorylation. This process produces a significant amount of ATP and carbon dioxide as a byproduct [20,28].

On the other hand, in anaerobic conditions, pyruvate can be converted to lactate, which can be used as a temporary energy source, and can accumulate in the muscle, contributing to fatigue [29,30].

(ii)Fatty Acids

Fatty acids are derived from dietary fats or stored triglycerides within the adipose tissue. They serve as an important source of energy, particularly during low- to moderate-intensity exercise or prolonged endurance activities. Mobilized from adipose tissue and transported into muscle cells mitochondria, fatty acids are broken down through a series of enzymatic reactions into acetyl-CoA molecules. Then, acetyl-CoA enters the citric acid cycle, generating ATP through oxidative phosphorylation. This process, called beta-oxidation, produces large amounts of ATP [31]. In metabolic myopathies, a sequence of events impairs energy production and utilization in muscle cells. Less glucose is oxidized due to deficiencies in enzymes or pathways involved in glycolysis; therefore, less intermediate molecules (i.e., fumarate, malate, oxaloacetate) are produced with consequent low reducing agent levels, such as NADH and FADH2, which can impact the electron transport chain’s ability to transfer electrons and generate a proton gradient, which is essential for ATP synthesis. Impaired ATP production can also affect the muscle’s ability to extract oxygen from the circulation and utilize it efficiently during exercise. This can contribute to fatigue and reduced exercise tolerance [32].

(iii)Others

In some specific conditions, muscle cells can use other energy substrates. During periods of prolonged fasting or a low-carbohydrate diet, the liver produces ketone bodies (acetoacetate and β-hydroxybutyrate) from fatty acids. Ketone bodies can be used by muscle cells as an alternative energy source, particularly during prolonged endurance exercise, thereby helping in sparing glycogen and potentially delaying the onset of muscle fatigue during prolonged or intense exercise by decreasing the lactic acid buildup during exercise [33].

Alternatively, during periods of fasting or when glycogen stores are depleted, certain amino acids, which are primarily used for protein synthesis, can be converted to pyruvate (e.g., alanine, serine, glycine, cysteine, threonine, tryptophan), or enter in the citric acid cycle at various intermediate points (e.g., alanine, aspartate, glutamate, phenylalanine and tyrosine, methionine, isoleucine) [34].

The utilization of glucose, fatty acids, and other energy substrates in muscle depends on factors such as exercise intensity, duration, and individual metabolic adaptations. Any impairments in the metabolism of these bioproducts affect muscle function, as it occurs in metabolic myopathies where depletion of muscle energy manifests as muscle fatigue.

#### 1.2.3. Enzymes Involved in Muscle Fatigue

Several enzymes play important roles in muscle fatigue and the metabolic processes associated with it. Among them, the following are distinguished:

##### Creatine Kinase (CK)

CK is involved in the conversion of creatine phosphate (CP) to generate ATP. CP acts as a readily available energy source for restoring ATP during intense muscle contractions. CK catalyzes the transfer of a phosphate group from CP to ADP, generating ATP. Changes in CK activity or the availability of CP can influence the rate of ATP regeneration and contribute to muscle fatigue [7].

##### Lactate Dehydrogenase (LDH)

LDH converts pyruvate, a byproduct of glucose metabolism, into lactate. During intense exercise, when oxygen availability is limited, glucose is metabolized through glycolysis to generate ATP. LDH helps facilitate this process by converting pyruvate into lactate. The accumulation of lactate can contribute to muscle fatigue and the sensation of myalgia or discomfort [35,36]. An ischemic forearm exercise test provides the rate of glycogenolysis [35,36]. Ischemia determines a switch from aerobic to anaerobic glycogenolysis. Lactate normally increases at least four-fold within 1–2 min of exercise; a diminished production occurs if glycogenolysis is defective or the exercise is on submaximal threshold.

##### Phosphofructokinase (PFK)

PFK is a key regulatory enzyme in the glycolytic pathway, which is responsible for the breakdown of glucose to generate ATP. PFK controls the rate-limiting step of glycolysis, converting fructose-6-phosphate to fructose-1,6-bisphosphate. Changes in PFK activity or regulation can affect the rate of glucose metabolism and impact the availability of ATP during muscle contractions [37].

##### Mitochondrial Enzymes

Enzymes involved in oxidative phosphorylation, the process by which ATP is produced in the mitochondria, also play a role in muscle fatigue. Enzymes such as cytochrome c oxidase, succinate dehydrogenase, and ATP synthase are involved in ATP production through the electron transport chain. Impairments in mitochondrial function or alterations in the activity of these enzymes can have an impact on ATP production and contribute to muscle fatigue [38,39].

#### 1.2.4. Ion Channels

The modulation activity of different ion channels affects the NMJ, and within the muscle fibers themselves the membrane potential, excitability, and calcium homeostasis.

In a resting muscle cell, the membrane potential is typically negative inside the cell due to the differential distribution of ions across the cell membrane. In detail, in the initial phase of an action potential, a rapid influx of Na^+^ through voltage-gated Na^+^ channels leads to depolarization of the membrane and triggers muscle contraction [40]. During muscle contraction, repeated action potentials are generated and propagated along the sarcolemma, leading to changes in ion gradients across the membrane. In particular, voltage-gated Na^+^ channels (VGSCs) and voltage-gated Ca^2+^ channels are involved in these processes [41,42,43]. High intracellular Na^+^ levels affect the efficiency of initiation, propagation, and frequency of potential action in muscle cells, and consequently reduce the excitability of muscle fibers [41].

A demodulated activity of other specific ion channels may contribute to muscle function impairment. During muscle contraction, potassium ions (K^+^) are released into the extracellular space. High levels of extracellular K^+^ induce depolarization of the sarcolemma and reduction in the excitability of muscle fibers, contributing to muscle fatigue by impairing the ability of the muscle to generate action potentials [44].

Ion channels also play a role in the recovery from muscle fatigue. After exercise, the restoration of ion gradients and the removal of accumulated ions are essential for muscle recovery. Ion channels involved in these processes, such as the Na^+^/K^+^ pump, are critical for returning the muscle to its resting state [40,45].

Muscle fatigue can indirectly result from a decrease in ATP availability due to metabolic disturbances, which impairs the ability of these pumps to maintain proper Ca^2+^ levels in the muscle cell.

## 2. Overview of Principal Metabolic Myopathies and Their Impact on Muscle Function

Metabolic myopathies are a heterogeneous group of rare inherited disorders, in which impairments in processes of storage, mobilization, and utilization of metabolic substrates result in the inefficient breakdown or utilization of energy sources within the muscle fibers. As a consequence, individuals with metabolic myopathies may experience muscle weakness, fatigue, and exercise intolerance. The concept of “experience” is sometimes very hard to be decoded by clinicians, considering the high rate of subjectivity in self-reported signs which may deviate from observable facts, particularly regarding the epidemiological relevance of apparently pathogenic mutations associated with asymptomatic or pauci-symptomatic clinical pictures (Figure 1).

### 2.1. Glycogen Storage Diseases (GSDs)

GSDs encompass a group of inherited autosomal recessive metabolic disorders caused by enzyme deficiencies involved in glycogen metabolism. Glycogen is a complex sugar molecule that serves as a stored form of glucose in the body. In GSDs, the ability to store or utilize glycogen is impaired, leading to the abnormal accumulation or depletion of glycogen in various tissues, particularly in liver and muscles. There are several types of GSDs, each associated with a specific enzyme deficiency.

In certain types of GSDs, the affected individuals may have difficulties breaking down glycogen into glucose, which is the main energy source for muscle cells. This occurs in GSD type II or GSD type V, named also Pompe disease (PD) or acid maltase deficiency disease and McArdle disease, respectively.

Pompe disease results from a deficiency or complete absence of the enzyme acid alpha-glucosidase (GAA), which breaks down glycogen into the lysosomes. The clinical phenotype of the disease can vary in severity and onset and primarily affects skeletal and cardiac muscle. The disease may present with early or late infantile onset (IOPD), adolescence, or adulthood; the clinical course is variable and progressive, characterized by muscle weakness, respiratory difficulties, fatigue, and motor impairment, and in some cases severe cardiac involvement. Involvement of the diaphragm and accessory respiratory muscles can induce sleep apneas with consequent poor blood oxygenation during the night hours and respiratory acidosis. Therefore, daytime sleepiness, fatigue, and fatigability occur [45].

In addition, the proteomic and lipidomic profile in PD patients shows signs of dysregulated autophagy, impaired calcium homeostasis, increased reactive oxygen species, inflammatory and immune response, and mitochondrial defects [46]. Oxidative stress is likely to play a role in autophagy induction, confirmed by deposition of lipofuscin into the muscle [47]. Upregulation of transforming growth factor beta 1 (TGF beta), tumor necrosis factor alpha (TNF alfa) [48], and matrix metallopeptidase 2 and 9 (MMP2 and MMP9) has been observed [49]. In severe clinical forms of IOPD, the expression of the gene encoding for the B-cell antigen receptor complex-associated protein alpha chain (CD79A), a marker of B cells, is elevated [50]. All these pathways are strictly interrelated with each other and affect the severity of the clinical spectrum.

In several human studies, the urine level of Glc4 has been found to correlate with IOPD progression and response to treatment [51,52]. Glc4 is derived from the intravascular degranulation of the glycogen released in circulation and corresponds to the degree of glycogen accumulation in skeletal muscles; it has been demonstrated to be a good progression marker of disease but, to date, the measurement in dried urine spot samples has low accuracy and requires complex methodologies [53].

More recently, it has been demonstrated in a preclinical model that several miRNAs can modulate the expression of genes involved in autophagy, muscle regeneration, and muscle atrophy pathways. In particular, miR-133a can influence muscle contractility and strength by modulating the expression of genes involved in muscle contraction and calcium handling; high levels of miR-133a correlate with phenotype severity mostly in infantile forms [54]. However, no specific biomarkers related to fatigue in PD have been identified.

The impaired production of 4-carbon tricarboxylic acid cycle intermediates (i.e., fumarate, malate, oxaloacetate) due to glycogenolysis failure reduce NADH and FADH2, which is fundamental to sustain the aerobic metabolism [54].

Both CK and LDH levels, aspartate aminotransferase (AST) and alanine aminotransferase (ALT), can be elevated in PD, but not specifically for the disease and it is inaccurate to predict the progression of muscle damage in PD [55].

Nevertheless, PD is defined as a primary metabolic disorder, and in infantile and juvenile forms, the involvement of brain cerebellum, anterior horn cells, and spinal cord is described, which leads to its consideration as a central component of fatigue [46].

In contrast to PD, in McArdle disease, the deficiency of the myophosphorylase enzyme induces the inability to mobilize glycogen in skeletal muscle cells for energy during exercise. As a consequence, muscle weakness and fatigue more typically present as exercise intolerance. Exercise-induced muscle cramps or myalgia are reported in affected patients when engaging in high-intensity exercise [56]. From a metabolic point of view, the so-called “second-wind” phenomenon explains what occurs in almost 50% of McArdle patients during prolonged exercise. Between 5 and 15 min of muscle work, an adaptation to the effort and a sudden improvement in fatigue after the initial period of difficulty or discomfort is present. During the first exhausting phase of muscle exercise, lactate and ammonia are produced; however, they are not directly related to the glycogenolysis process. Similarly, phosphorylated intermediates of the gluconeogenesis, such as phosphoenolpyruvate carboxykinase or hexose phosphate derived from blood circulating glucose are a sufficient energy supply and induce lactate production. In addition, some amino acids, such as alanine, could be utilized to provide energy in the initial stage of muscle exercise. Ammonia, generally increasing at least five-fold within 2–5 min of exercise, is derived from the degradation of adenosine monophosphate (AMP) into inosine monophosphate (IMP); it acts as way to restore ATP in depleted conditions, such as during exercise in McArdle disease [57]. Therefore, there is a lack of elevation in blood lactate and excess of ammonia production during an ischemic handgrip exercise, excess of glycogen, and deficit of myophosphorylase activity in the muscle biopsy. Also, the myophosphorylase expression test in white blood cells can be considered complementarily diagnostic [58]. On the other hand, inducing the re-expression of myophosphorylase in a murine animal model causes a decrease in glycogen storage in the muscle with consequent increase in strength contraction and decrease in fatigability [59].

Santacatterina et al. investigated the expression of energy metabolism proteins in muscle biopsy samples from a group of patients affected by different neuromuscular diseases. Reverse phase protein microarray (RPMA) quantification tests showed lower levels of pyruvate dehydrogenase E1 alpha 1 subunit (PDH E1alpha), cytochrome c oxidase subunit I (COX1), and human lactate dehydrogenase A (LDH-A) in patients with GSDV compared to healthy controls [60].

A significant downregulation of sarcoplasmic reticulum Ca^2+^ ATPase 1 (SERCA1) and glycogen synthase (GS) of 75% and 50%, respectively was found in GSDV compared to controls, hypothesizing that lower levels of SERCA1 can impact calcium transport in type II muscle fibers, thereby leading to early onset fatigue [61].

Poly (ADP-ribose) polymerase 1 (PARP1) induces the adaptive response of muscle tissue to exercise stress, potentially influencing muscle fiber type and mitochondrial function [62]. In muscular diseases, PARP1 activity can be upregulated, contributing to tissue damage and inflammation through the generation of poly(ADP-ribose) polymers and depletion of cellular energy stores (NAD+ and ATP). This has been demonstrated in a mouse model of GSDV [63]. Moreover, the same authors found that adaptations to endurance-exercise training involve the mitochondrial structure (such as the proton-transporting ATP synthase complex and respiratory electron transport chain) [63].

Another important protein is MAPK12, a protein kinase belonging to the mitogen-activated protein kinase family, which regulates cellular responses to stress and inflammation, as well as energetic homeostasis. Although it is not a primary regulator of energy metabolism during exercise, it seems to facilitate glucose transport and improve mitochondrial function [64].

In other glycogen storage diseases, such as type III (GSDIII, Cori-Forbes disease) or VIII and IX, skeletal muscle damage is related to other specific deficit enzymes involved in glycogen breakdown, respectively amylo-alpha-1,6-glucosidase or 4-alpha-glucanotransferase (AGL) and liver phosphorylase b kinase (PhK) enzyme. In a more rare variant of GSDVIII, a prominent muscle involvement due to muscle PhK deficiency which determines muscle exercise intolerance and fatigability, is described [55].

In GDSIII, the inhibitory subunit of Troponin (TnI) presents not only a myocardial isoform, but also two different skeletal muscle isoforms (sTNI) which are not present in myocardial tissue. Therefore, an increase in plasma sTnI is an expression of muscle damage [65].

### 2.2. Fatty Acid Oxidation Disorders (FAODs)

Fatty acids are a type of lipid that comprise carbon, hydrogen, and oxygen atoms, synthesized by the body or taken with diet, which serve as a major energy source in the body, especially during prolonged exercise or fasting. In mitochondria, fatty acids undergo a process called beta-oxidation with consequent splitting into acetyl-CoA molecules. Acetyl-CoA enters the Krebs cycle and generates ATP. With the dietary intake, excess fatty acids are stored as triglycerides in adipocytes and mobilized to be used for energy production in conditions of energy deficit. In addition, fatty acids are essential structural components of cell membranes, contributing to membrane fluidity and playing a role in cell signaling and networking [66,67].

In some genetic clinical conditions, defects in the enzymes are responsible for the breakdown of fatty acids, or in their transport, proteins induce fuel source depletion [68]. More rarely, toxic causes or an unbalanced diet are responsible for this metabolic impairment. In any case, glucose stores can be depleted, leading to an increased reliance on fatty acid oxidation. However, due to the underlying enzyme deficiency, the cells are unable to effectively utilize fatty acids for energy production. In addition, in some cases, this metabolic imbalance leads to the potential accumulation of toxic byproducts which in turn, can further affect cell metabolism [66].

Enzyme deficiencies in FAODs include deficiencies of medium-chain acyl-CoA dehydrogenase (MCAD), long-chain acyl-CoA dehydrogenase (LCAD), very-long-chain acyl-CoA dehydrogenase (VLCAD), etc. However, myopathic involvement is not predominant in all forms. Carnitine palmitoyltransferase (CPT) is one of the enzymes carrying long-chain fatty acids across the mitochondrial membrane, and its deficiency can be classified in three main biochemical types, defined as CPT-I, which is the most common form but not primarily related to myopathic condition, CPT-II or CPT-III deficiency [69].

CPT-II transfers fatty acids from the carnitine molecule inside the mitochondria to the mitochondrial matrix for energy production. CPT-III is an enzyme involved in the regulation of CPT-I activity and its deficiency results in a less severe clinical form than CPT-I and CPT-II deficiencies [70,71].

CPT-II deficiencies are the most common myopathic disorders caused by the alteration of lipid metabolism [71].

The onset of FAODs can occur in infancy, childhood, or even adulthood, and can vary, depending on the specific enzyme deficiency and its severity.

Usually, patients affected by lipid myopathy experience muscle weakness and fatigue, which often get worse during prolonged exercise or periods of fasting, or are triggered by other stressors that require increased energy production such as illness conditions [72].

In the case of structured myopathy, CK elevation ranges from two to five times the normal value, while in the case of acute onset with rhabdomyolysis and myoglobinuria, the value can increase to several thousands of units [69].

In CPT-II-related myopathy, which presents as recurrent attacks of myalgia, cramps, muscle stiffness or tenderness, transient weakness and rhabdomyolysis, muscle exercise is an important trigger. Plasma-free carnitines are normal, while C16 and C18 acylcarnitine species increase [50].

Alteration of acylcarnitine profile, with predominant C14:1 chain, is present also in very-long-chain acyl-CoA dehydrogenase (VLCAD) in which rhabdomyolysis episodes are induced by stressor triggers, such as muscle exertion. VLCAD function is related to acyl-CoA dehydrogenase 9 in initializing fatty acid oxidation, in particular ACAD9 participates in mitochondrial complex I assembly [73]. Most patients with ACAD9 deficiency experience exercise intolerance and peripheral lactic acidosis is variably present [74].

In the milder late-onset myo-neuropathic form related to mitochondrial trifunctional protein (MTP) dysfunction, recurrent rhabdomyolysis is often associated with respiratory failure and signs of sensorimotor axonal neuropathy. MTP plays a pivotal role in the breakdown of long-chain fatty acids for energy production within the mitochondria, in particular it activates long-chain enoyl-CoA hydratase (ECH), long-chain L-3-hydroxyacyl-CoA dehydrogenase (LCHAD), and long-chain 3-ketoacyl-CoA thiolase (LCKAT) [75]. From the biochemical profile, levels of total carnitine are decreased and different long chain fatty acids appear variably elevated during episodes of rhabdomyolysis [69].

In primary carnitine deficiency (PCD), a recessive gene mutation in *Solute Carrier Family 22 Member 5 (SLC22A5)* causes a defect in the function of organic cation/carnitine transporter 2 (OCTN2), leading to a decrease in renal reabsorption of carnitine [69]. Thus, carnitine concentration in plasma and tissue and its uptake in lymphocytes and fibroblasts are reduced. An amount of <5% of normal concentration of plasma carnitine with normal esterified fractions and high carnitine loss in urine are pathognomonic [76]. On the contrary, in some neutral lipid storage diseases (NLSDs) in which myopathy can occur, such as the form related to *Patatin Like Phospholipase Domain Containing 2 (PNPLA2)* gene mutation, the profile of acylcarnitine and carnitine is typically normal [69].

Multiple acyl-CoA dehydrogenase deficiency (MADD) can present with a broad phenotypic spectrum [77].The adult onset form is typically characterized by exercise intolerance, and in some cases with fluctuations [78]. The low concentration of total carnitine and elevated levels of acylcarnitines (C4–C18) are found in plasma; organic dicarboxylic acids and acylglycine derivates are detected in urines.

In isolated forms of coenzyme Q10 deficiency (caused by mutations in gene for electron transfer flavoprotein dehydrogenase, ETFDH), CK elevation is also present. In muscle biopsy, lipid accumulation with occasional ragged red fibers (RRFs) and COX negative fibers are frequently seen since reduced activities of mitochondrial complexes and decreased CoQ10 are reported [79].

In histone deacetylase 11 (HDAC11) total knockout (KO) mice (HDAC11^−^/^−^ or KO mice) model, loss of HDAC11 increased the count of oxidative muscle fibers by facilitating a transition of muscle fibers from glycolytic to oxidative pathway. Furthermore, HDAC11 was localized in muscle mitochondria where, by the activation of protein kinase-acetyl-CoA carboxylase pathway mediated by AMP and reducing acylcarnitine levels, fatty acid beta-oxidation is enhanced. Overall, fatigue resistance and muscle strength were enhanced [65].

Overexpression of sarcolipin (SLN), a regulator of sarco(endo)plasmic reticulum calcium ATPase (SERCA) in skeletal muscle, in an animal model, improves the muscle contractile properties, endurance exercise capacity, fatigue and metabolic rate by increasing CPT-I and lactate dehydrogenase expression and Ca^2+^ entry in the cytosol, thereby enhancing fatty acid metabolism and glycolytic capacity [80].

### 2.3. Mitochondrial Myopathies

Mitochondria are highly dynamic, small, double-membraned organelles essential for eukaryotic life. Among their main roles, they intervene in the oxidative phosphorylation by transfer of high-energy electrons through transport chains, generating an electrochemical gradient that drives protons across the membrane through ATP synthase enzyme. This chemiosmotic process culminates with the synthesis of adenosine triphosphate (ATP), the main energy source for cellular activities [81].

Furthermore, mitochondria are involved in several metabolic pathways. They participate in the breakdown of fatty acids through beta-oxidation, the oxidation of amino acids, and the metabolism of carbohydrates, such as glucose. These processes provide substrates and intermediates for ATP production, biosynthesis, and other cellular functions.

In addition, mitochondria regulate the uptake and storage of Ca^2+^, which act as a homeostatic regulator of cellular signaling processes and is crucial for proper muscle function. Strictly related to that, mitochondria can adapt their function in response to exercise training. In particular, endurance exercise training can increase the number and size of mitochondria in muscle cells, enhancing the capacity for ATP production and improving muscle endurance [82].

It is therefore intuitive that mitochondria are abundant in cells with high energy demands, such as muscle cells. As a consequence, defects in mitochondrial activity affect muscle function and integrity, figuring out conditions of mitochondrial myopathies, due to several mutations in genes involved in mitochondrial functions. The symptoms of mitochondrial myopathies depend on the specific mitochondrial defect and muscles affected and include different extents of muscle weakness, exercise intolerance, myalgia, cramps, and fatigue, which are experienced by affected individuals particularly during aerobic activities [83].

The most investigated markers related to mitochondrial function and dysfunction were carnitine, genetic profiles regarding metabolism, energy production, protein transport, and mitochondrial morphology. Coenzyme Q10 (CoQ10), or ubiquinone, is the most commonly investigated mitochondrial enzyme. Low levels of CoQ10 were consistently associated with fatigue [39]. CoQ10 is a lipophilic inner-membrane co-factor involved in the production of energy within cells and serves as a powerful antioxidant. Its role is to vehicle electrons from complex I and complex II to complex III. While CoQ10 does not directly participate in the breakdown of fatty acids, it plays a crucial indirect role by facilitating the transport of electrons during the oxidation of fatty acids. CoQ10 also exerts an antioxidant activity [84]. In patients affected by mitochondrial disorders, administration of CoQ10 improves motor performance on aerobic cycle exercise and post-exercise blood lactate, but no effect on strength or resting blood lactate was found [85].

Nrf2 transcription factor is involved in promoting mitochondrial function, including energy production, reactive oxygen species generation, calcium signaling, and cell death induction [86].

During a 12-week trial treatment with omaveloxolone, a reduction in lactate levels during submaximal exercise was observed. Nevertheless, there was no difference in the clinical outcomes [87].

Mitochondrial biogenesis depends also on a pathway mediated by PGC-1α, which is in turn activated by NAD+ (nicotinamide adenine dinucleotide), cofactor forSIRT1. NAD+ deficiency is observed in mitochondrial patients, and supplementation with NAD+ precursors increases the mitochondrial metabolism [46]. In an open-label study conducted on patients who were taking niacin (nicotinic acid), blood and muscle NAD+ reached control levels [88].

## 3. Conclusions and Future Perspectives

Fatigue and exercise intolerance are very common and multifaceted symptoms in metabolic myopathies, implying the existence of some extent of functional and/or structural muscle impairment.

In general, the impact of metabolic myopathies on muscle function can vary, depending on the specific disorder and its severity. Understanding how several molecules interact in different pathways and how their availability is affected at rest and during exercise provides insights into the limitations of muscle function and recovery due to unmet energy demands.

To date, we have biomarkers that collectively provide insights into different aspects of muscle fatigue such as metabolic changes, oxidative stress, inflammation, and muscle damage. The use of high-throughput assays, such as next-generation sequencing and mass spectrometry, or advanced omics technologies, including genomics, proteomics, and metabolomics, have significantly accelerated the discovery and validation of potential muscle fatigue biomarkers, enabling the simultaneous examination of thousands of molecules and their changes during muscle exertion, thereby ensuring the validation and refinement of the utility of several biomarkers in different contexts. However, even excluding cost-effectiveness issues, the access to specialized laboratory techniques or diagnostic tools in clinical practice is very limited considering the high prevalence and complexity of the fatigue phenomenon. Furthermore, whenever possible, noninvasive or minimally invasive methods, such as blood, urine, or saliva samples, should be preferred. Finally, studying muscle fatigue biomarkers contributes to a deeper understanding of its underlying mechanisms, facilitating the development of targeted and personalized interventions and therapies, such as adaptation of training programs to improve muscle metabolic response.

## Figures and Tables

**Figure 1 biomolecules-14-00050-f001:**
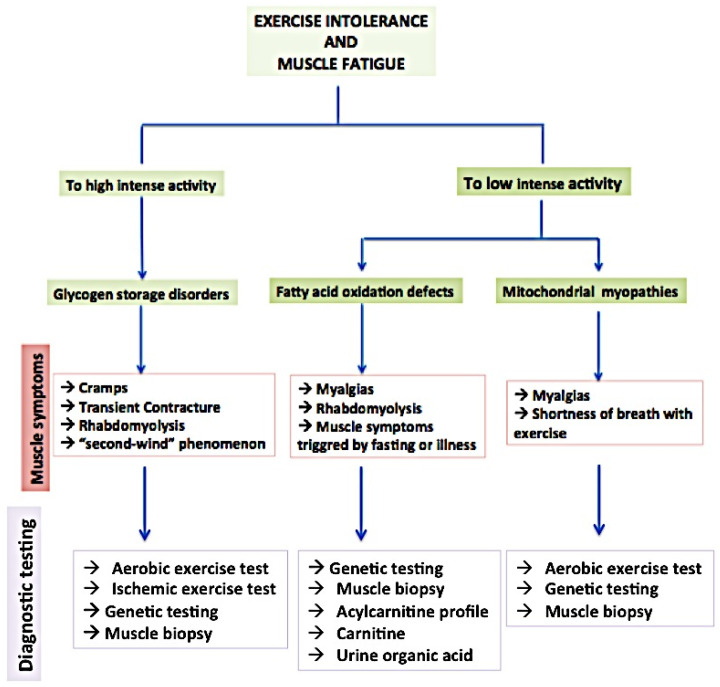
Clinical red flags and diagnostic algorithm for metabolic myopathies.

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
