# Peer review of "Biomolecules of Muscle Fatigue in Metabolic Myopathies"

_biomolecules, 2023, doi:10.3390/biom14010050_

Round 1

Reviewer 1 Report

Comments and Suggestions for Authors

The review of Schirinzi et al. deals with the key biomolecules involved in muscle fatigue in metabolic myopathies. In my opinion the review is quite confusing and not very understandable. It would have to be heavily edited before to be considered for the publication.

Some major suggestions:

- the review does not contain a figure and/or table that would greatly aid the reader in understanding. One or more figures/tables could be added to the paper

- paragraph 1.2.1 is too divided, in my opinion it should be less of a list of things and more discursive

Author Response

Thank you for your suggestions. The paper has been edited, a schematic diagram has been added  and paragraphs division riviseted also in accordance with Editor suggestions.

Reviewer 2 Report

Comments and Suggestions for Authors

While the topic of the paper may be of interest to the readership, it is very difficult to get through as written. There are significant grammatical errors throughout. Additionally, the paper currently lacks structure, adding to the difficulty in readability. 

The authors are encouraged to work with a native English speaker to address the grammatical errors. 

Comments on the Quality of English Language

The authors are encouraged to work with a native English speaker to address the grammatical errors. 

Author Response

Thank you for your comments. English language has been revised and paper structure modified also in accordance with Editor suggestions. 

Reviewer 3 Report

Comments and Suggestions for Authors

I appreciate the opportunity to review this excellent review article submitted for publication in Biomolecules. While the phenomenon of muscle fatigue is well-known from a clinical perspective in the practice of neurologists specializing in Neuromuscular Diseases, it is not a manifestation restricted to this group of diseases. There is also significant variation in the origin of this symptom across different clinical conditions, which may even reflect clinical dysfunctions of non-neurological origin. The authors have focused on the theme of metabolic myopathies. The text is very up-to-date and has an excellent foundation of reviewed literature, being one of the few that has sought to deepen knowledge specifically in the aspect of muscle fatigue. My only suggestion to the authors would be the addition of a figure or a schematic representation of the dysfunctions and pathophysiology observed in muscle fatigue in the major metabolic myopathies.

Comments on the Quality of English Language

A significant improvement in the quality of the manuscript's text will be achieved after a native English speaker reviews the written content. In several paragraphs (i.e., line 403), there are errors in relatively simple structures, such as the placement of punctuation with commas between the subject and verb, which need to be reviewed. These are patterns that minimally compromise the quality of this excellent review manuscript.

Author Response

Thank you for your positive feedback and suggestions. Grammar errors have been reviewed and papaer structure better organized. A schematic figure “flow-chart type”has been added.

Round 2

Reviewer 1 Report

Comments and Suggestions for Authors

The authors modified the paper according to the referee' suggestions and in my opinion is now suitable for pubblication.

Author Response

Dear Reviewer,

thank for your feedback.

Reviewer 2 Report

Comments and Suggestions for Authors

Unfortunately, the manuscript in its present form is too long and unfocused. It reads more like a series of book chapters than a manuscript focusing on the biomolecules of muscle fatigue.

Comments on the Quality of English Language

There are still several grammatical errors and issues with flow. The authors are encouraged to work on cutting ~60% of the material and extraneous words.

One of the main problems with this article is the lack of focus. In other words, many sections are completely unnecessary to tell the story of muscle fatigue. For example, at the beginning of the article, a lengthy discussion of muscle contraction describes excitation-contraction coupling. If the purpose of the manuscript is to describe the biomolecules that contribute to muscle fatigue, then focusing on factors that contribute to declines in muscle force production and energy metabolism should be the priority.

Other sections of the article also need to be edited for content to have extraneous material (and phrases) removed.

Author Response

Dear Reviewer,

taking into account your comments, the text has been newly edited.
